# 3D collagen high-throughput screen identifies drugs that induce epithelial polarity and enhance chemotherapy response in colorectal cancer
Sarah J. Harmych [1,2], Thomas P. Hasaka[3], Chelsie K. Sievers[1], Seung Woo Kang [2,4,5], Marisol A. Ramirez[6,7], Vivian Truong Jones[1,8], Zhiguo Zhao[6], Oleg Kovtun [9], Claudia C. Wahoski[1,10], Qi Liu[6,7], Ken S. Lau [2,4,5,11,12], Robert J. Coffey [1,4], Joshua A. Bauer [3,12,13] & Bhuminder Singh [1,2,4,10,12] ✉

Loss of polarity is a hallmark of cancer, and the related epithelial-to-mesenchymal transition (EMT) phenotype impacts prognosis and therapy outcomes, particularly in colorectal cancer (CRC). However, the mechanisms and drugs that impact EMT-related morphological changes are understudied, due to the complete failure of typical live/dead 2D high-throughput screens to capture morphology or the lack of robustness of 3D screens. We designed a high-throughput screen using 3D type I collagen cultures of CRC cells to assess morphological changes in colonies and identified several FDA-approved drugs that re-epithelialize CRC colonies. One of these drugs, azithromycin, increased colony circularity, enhanced E-cadherin membrane localization and ZO-1 localization to tight junctions, caused transcriptomic changes consistent with downregulation of EMT, and elevated sensitivity to the chemotherapeutic, irinotecan. A retrospective analysis of patient data demonstrated that the use of azithromycin in patients undergoing treatment for CRC with irinotecan had improved the 5 year survival compared to the chemotherapy alone. These results highlight the importance of morphological screens to identify novel drug candidates and synergistic mechanisms.

Epithelial cells line a majority of body cavities where they organize into polarized monolayers. Most cancers arise from the epithelium, and loss of epithelial polarity is essential in the development of cancer, facilitating hyperproliferation, multi-layering, and metastasis[1]. Cell or tissue polarity is considered a non-canonical tumor suppressor, serving to regulate proliferation, survival, and apoptosis of epithelial cells[2]. The epithelial-to-mesenchymal transition (EMT) requires the loss of apicobasal polarity and the establishment of front-to-rear polarity, causing a significant change in cellular shape and morphology of colonies[3]. Given that EMT has also been implicated as a mechanism of therapy resistance[4], assessing morphological

changes associated with EMT could prove an effective method in identifying drugs that enhance current treatments.

Features of epithelial polarity are better recapitulated in vitro by growing cells in 3D extracellular support matrices, like Matrigel, hydrogel, and collagen, than in 2D plastic cultures[5–7]. For example, when grown in type I collagen, the human colorectal cancer (CRC) cell line, HCA-7 forms primarily unilamellar colonies with intact apicobasal polarity (cystic colonies) and less frequent colonies made of irregular masses of cells (spiky colonies)[8]. Cell lines generated from cystic and spiky colonies, termed CC and SC cells respectively, look indistinguishable when grown on plastic but

[1]Department of Medicine, Vanderbilt University Medical Center, Nashville, TN, USA. [2]Department of Cell and Developmental Biology, Vanderbilt University, Nashville, TN, USA. [3]Vanderbilt Institute of Chemical Biology, High-Throughput Screening Facility, Vanderbilt University, Nashville, TN, USA. [4]Epithelial Biology Center, Vanderbilt University Medical Center, Nashville, TN, USA. [5]Center for Computational Systems Biology, Vanderbilt University, Nashville, TN, USA. [6]Department of Biostatistics, Vanderbilt University Medical Center, Nashville, TN, 37232, USA. [7]Center for Quantitative Sciences, Vanderbilt University Medical Center, Nashville, TN, USA. [8]Department of Pharmacology, Vanderbilt University, Nashville, TN, USA. [9]Department of Chemistry, Vanderbilt University, Nashville, TN, USA. [10]Program in Cancer Biology, Vanderbilt University, Nashville, TN, USA. [11]Department of Surgery, Vanderbilt University Medical Center, Nashville, TN, USA. [12]Vanderbilt-Cancer Ingram Center, Vanderbilt University Medical Center, Nashville, TN, USA. [13]Department of Biochemistry, Vanderbilt University School of Medicine, Nashville, TN, USA. ✉e-mail: bhuminder.singh@vumc.org

have different morphological and functional properties in 3D[9]. A notable difference between the cell lines is their sensitivity to the Epidermal Growth Factor Receptor (EGFR) neutralizing antibody cetuximab[9]. CC cell colony formation in type I collagen was significantly decreased with cetuximab treatment. In contrast, cetuximab treatment had little impact on SC colony formation[9]. Resistance of SC cells to cetuximab was determined to be due to increased activation of the receptor tyrosine kinases MET and RON, which work in parallel to the EGFR pathway blocked by cetuximab[9]. Resistance to cetuximab was induced in CC cells through the overexpression of the MET ligand, HGF. CC cells overexpressing HGF (CC-HGF) formed spiky colonies like those of SC cells rather than cystic colonies typical of CC cells. Like SC cells, treatment of CC-HGF cells with a MET/RON inhibitor, restored sensitivity to cetuximab and their cystic morphology[9–11].

As demonstrated by the study of CC and SC cells and other past studies, the morphology of 3D cancer cell colonies can reveal critical insights into functional characteristics, including response to drug treatment[8,9]. Numerous studies over the last five decades have shown the translational relevance of 3D cultures and the limitations of traditional 2D culturing methods[7]. Despite this, 2D culturing has remained the standard in the high-throughput screening field. Technical challenges, including the handling and plating of hydrogel-based artificial extracellular matrices and designing image analysis protocols to assess morphological changes in 3D cultures, have been hurdles in the widespread use of these cultures in drug screening[7,12].

In this study, we established a collagen-based high-throughput drug screen and identified several FDA-approved drugs which induce morphological changes in CRC colonies. The method developed for this study allowed for the identification of multiple distinct morphological phenotypes utilizing a variety of morphological characteristics. Furthermore, drugs identified in the screen for inducing an epithelial morphology were also able to enhance the response of colonies to chemotherapy treatment. The results of this study demonstrate the effectiveness of 3D morphological high-throughput screens in the identification of drugs which can be used to enhance current therapies for CRC.

## Results

### 3D high-throughput drug screen identifies distinct morphological classes of SC colonies

Our standard 12/24-well format 3D type I collagen culture (see methods) was adapted for use in a 384-well format with automated liquid handling equipment (Fig. 1A). After growing cells in 3D with a library of 1059 FDA-approved compounds for 8 days, colonies were stained with Calcein AM and confocal images were acquired and quantified for several parameters. The FDA-approved drug library contains drugs with a wide variety of functional targets, which are outlined in Fig. 1B. Measurements of morphological characteristics for all drugs screened are available as supplemental data (Supplementary Data 1) and raw images from the screen are available in a repository[13]. All these characteristics were measured and

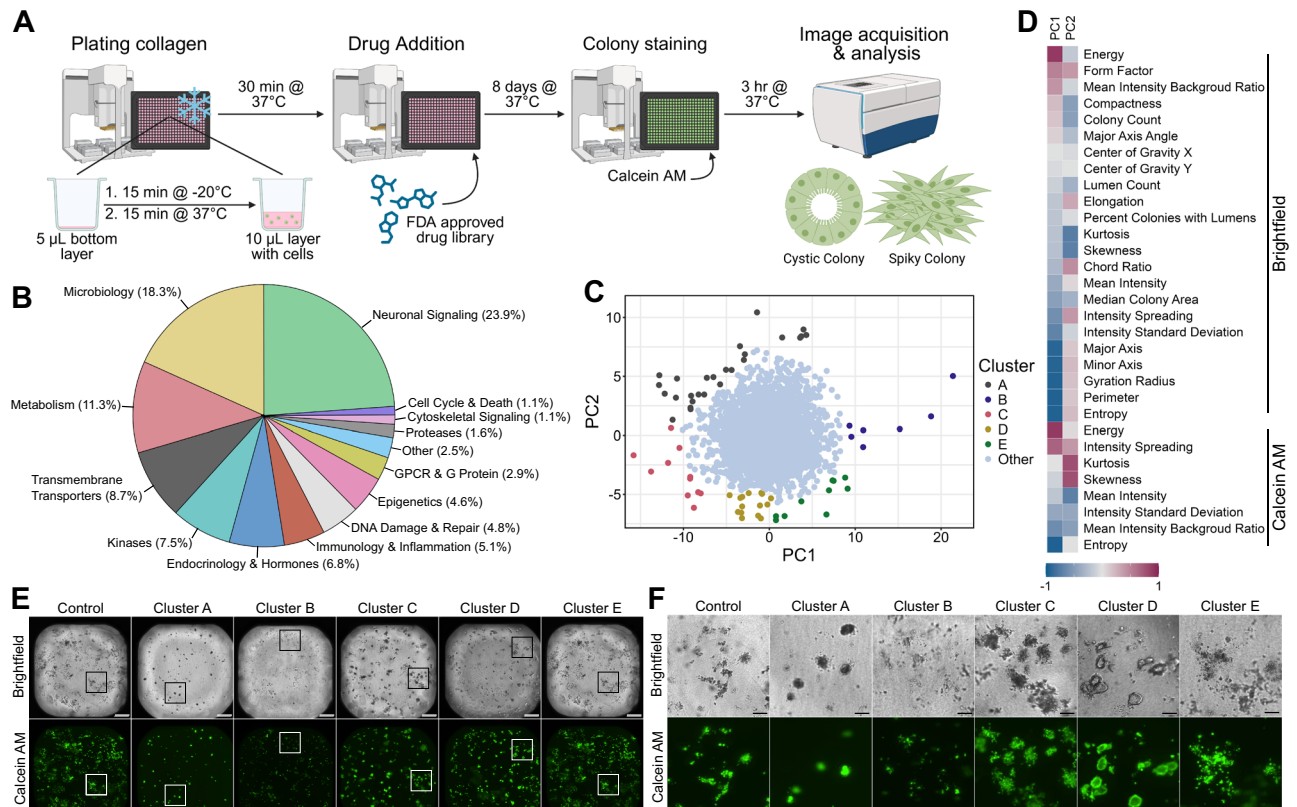

**Fig. 1 | High-throughput drug screen using 3D type I collagen cultures identifies morphological clusters. A** Schematic of plating, treatment, and analysis of drug screen. Chilled 384-well plates were stamped with a 5 μL bottom layer of type I collagen and allowed to solidify. 1000 SC cells in 10 μL collagen were then added and allowed to solidify. Next, 1059 compounds from an FDA-approved drug library were added at three concentrations and incubated for 8 days. Wells were then stained with Calcein AM, imaged, and analyzed to assess colony morphology. Created in BioRender. (Harmych, S. (2025) https://BioRender.com/4zfblch). **B** Pie graph depicting functional targets of drugs tested during screen. **C** Principal component analysis (PCA) plot of morphological characteristics of colonies from high-throughput drug screen. Morphological characteristics were obtained using the InCarta software. Each dot represents a compound-treated well from the screen. Distinct Clusters (**A–E**) separating from the central cluster were visually assessed and shown on the PCA plot. **D** Heatmap of PC loadings of each variable on PC1 and PC2. Brightfield and Calcein AM indicate which image was used by InCarta software to generate the value. **E** Representative whole well images of morphological clusters identified in the screen. Boxes correspond to images shared in Fig. 1F. Brightfield and fluorescent (Calcein AM) images taken with ImageXpress confocal HT.ai automated high-content imaging system at 4x magnification. (Scale bars: 500 μm). **F** Insets of representative wells shown in Fig. 1E. (Scale bars: 100 μm).

quantified automatically by the InCarta software, which utilizes artificial intelligence to analyze images. Pair-wise Pearson correlation coefficients were calculated between each variable (Supplementary Fig. 1A). Principal component analysis (PCA) was conducted subsequently, where 10 principal components (PCs) could describe over 95% of data variance (Supplementary Fig. 1B). Using this PCA, we first sought to determine if there were any discernable changes in the morphology of SC colonies when treated with the drugs in the screen. A scatterplot mapping of PC1 and PC2 revealed five distinct clusters of wells separate from the main cluster of wells along both the PC1 and PC2 axes (Fig. 1C). The PC loadings heatmap indicated energy and entropy were the main contributors of PC1, whereas kurtosis and skewness contributed the most to PC2 (Fig. 1D). Clusters were identified by visual assessment of wells that had separated from the central cluster of wells. The central cluster contained untreated control wells and drugs in the screen which had little to no effect on colony morphology compared to control wells. Drugs which largely had a cell death effect, as determined by decreased colony count, were unsuitable for colony morphology assessment and were removed prior to conducting the PCA.

The five clusters identified during PCA analysis each exhibited a distinct morphology (Fig. 1E, F). Drugs used to treat wells from Clusters A and B appeared to induce some cell death, as these Clusters showed fewer and smaller colonies than other wells. Colonies in Cluster A became small and dense while the colonies in Cluster B became sparse. Both Cluster C and Cluster E generated large colonies without a centralized lumen. Colonies in Cluster C had a dense center with spiky protrusions while those in Cluster E were disorganized and spread out, lacking the density seen in Cluster C. Colonies in Cluster D were also large and some had formed a central lumen suggesting re-epithelialization may have taken place. Overall, these results indicate that several distinct colony morphologies induced by drug incubation can be monitored and quantified in this unsupervised high-throughput screening format.

## Median colony area and percent colonies with lumens used to identify drugs re-circularizing SC colonies

We next investigated which morphological characteristics could be used to identify our morphology of interest, a spiky to cystic conversion of colonies. The integrin β1 antibody (P4G11), which has previously been shown to re-

epithelialize SC colonies in collagen[14,15] was used as a positive or "cystic" control for this screen. Including P4G11-treated wells, pair-wise Pearson correlation analysis (Supplementary Fig. 2A) and PCA were conducted again, where 10 PCs could describe over 95% of data variance (Supplementary Fig. 2B). A scatterplot mapping PC1 and PC2 revealed the positive control wells clearly separated from the other wells along the PC1 axis (Fig. 2A). Interestingly, PC loadings heatmap revealed several variables – such as minor axes, entropy, and related texture features – as major contributors, underscoring the unbiased, unsupervised strength of our automated quantification approach (Fig. 2B). Notably, some of these parameters also earlier showed relevance across the full screen (Fig. 1C). While these parameters may prove valuable following further validation, particularly for assessing unknown drug effects, our primary focus in tracking reversal to epithelial phenotype was to ensure visual confirmation of morphological conversion, minimizing false positives.

To that end, we prioritized features that were both visually intuitive and functionally meaningful. Metrics such as average and median colony area, perimeter, and lumen count—derived from InCarta software—aligned well with the expected formation of larger, smoother colonies with hollow lumens (Fig. 2B). Among these, we then focused on "median colony area" and "percent colonies with lumens" that were quantified through MetaXpress image analysis software. Plotting of the untreated spiky control wells (DMSO) and the P4G11-treated cystic control wells based on both median colony area and percent colonies with lumens showed that the two controls formed distinct clusters (Fig. 2C). Cystic control wells had significantly higher median colony area (Fig. 2D) and percent colonies with lumens (Fig. 2E) than the spiky control wells. Some lumens in the spiky controls likely reflect either true lumen-containing colonies, seen in about 22% of SC colonies[9], or artifacts due to a small hole between colonies that was not filtered out during image analysis. While the InCarta software was used for unsupervised clustering (Figs. 1C and 2B), subsequent quantification (Fig. 2C–E) was performed using MetaXpress analysis, which more accurately identified colony areas. Representative masks of colonies and lumens (Fig. 2F) were generated using Calcein AM staining and MetaXpress software for these analyses. Thus, the parameters colony size and percent colonies with lumens, identified from our positive control for cystic morphology

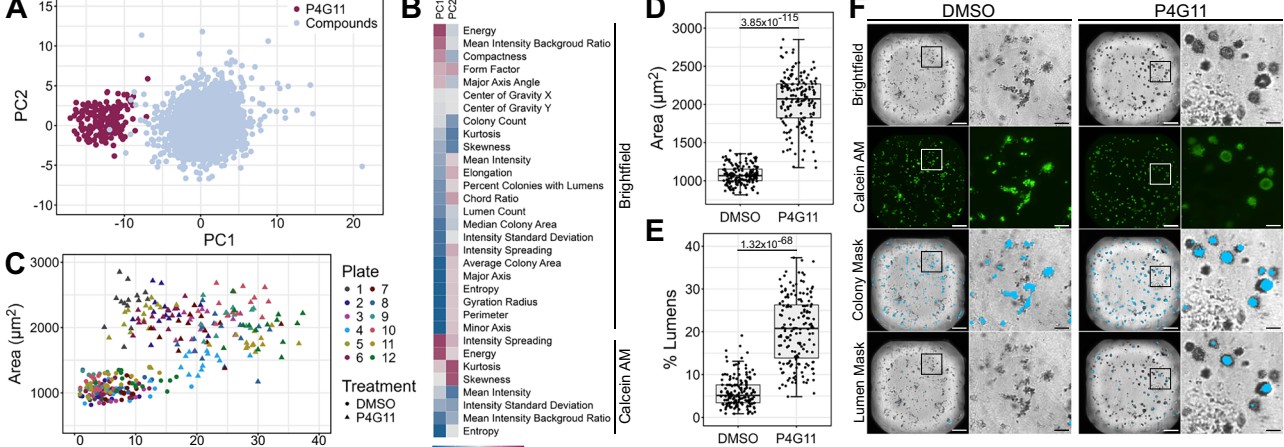

**Fig. 2 | Median colony area and percent colonies with lumens identify re-epithelialized colonies. A** Principal component analysis (PCA) plot of colony morphological characteristics from high-throughput drug screen. Each dot represents a well from the screen. Highlighted dots indicate wells treated with the positive control, P4G11. **B** Heatmap of PC loadings of each variable on PC1 and PC2. Brightfield and Calcein AM indicate which image was used by InCarta software to generate the value. **C** Comparison of DMSO and P4G11 control wells using median colony area and percent colonies with lumens determined using MetaXpress software. Each dot represents a well from the screen. **D-E** Comparison of (**D**) median

colony area and (**E**) percent colonies with lumens for control wells. Box and whisker plots denote median (center line), interquartile range (box), and minimum and maximum (whiskers). Wilcoxon rank-sum test, *p*-values indicated. **F** Representative brightfield and fluorescent (Calcein AM) confocal images of DMSO and P4G11 control wells from screen. Left panel depicts whole well images (Scale bars: 500 μm). Right panel depicts higher magnification of indicated section of the well (Scale bars: 100 μm). Images taken with ImageXpress confocal HT.ai automated high-content imaging system at 4x magnification. Colony and lumen masks are overlaid on brightfield images and were generated using MetaXpress image analysis software.

were then used to identify hits from our drug screen that re-circularize SC cell colonies.

## Antibiotics azithromycin, clindamycin, and linezolid induced re-circularization of SC colonies in screen

Hits of interest from the screen were identified based on median colony area (Fig. 3A) and percent colonies with lumens (Fig. 3B). Both morphological characteristics were assessed using a control-based measurement and a control-independent measurement[16]. The control-based measurement, fold-change, was based on the DMSO control wells of the same plate. The control-independent measurement, B-score, accounts for both plate-to-plate variability and positional variability within a plate. Fold-change and B-score were used to plot each well and then the top 5% for each was calculated (Fig. 3A, B). This generated an upper quadrant on the plot containing wells within the top 5% for both fold-change and B-score. Wells that fell within this upper quadrant for both median colony area and percent colonies with lumens were considered hits (Fig. 3C). Ten wells in the screen met the criteria to be considered a hit. Based on visual assessment, two of these wells were determined to be false positives. The remaining eight wells corresponded to six different drugs, four of which were antibiotics.

The four antibiotics identified as hits during the screen were azithromycin, clindamycin, linezolid, and methacycline hydrochloride. While these hits belong to different antibiotic classes, all four bind to the bacterial ribosome, inhibiting protein synthesis and ultimately bacterial replication. The other two drugs identified as hits were ruxolitinib, a JAK inhibitor, and diminazene aceturate, an antiparasitic. When the three concentrations of each drug were compared to their corresponding plate from the screen, at least one well fell above the upper interquartile range for both median colony area and percent colonies with lumens (Fig. 3D, E, Supplementary Fig. 4A, B). Circular colonies with distinct lumens can be observed in at least one of the concentrations for each drug (Fig. 3E, F, Supplementary Figs. 3B, 4B–D). Wells from each of these hits also clustered together when highlighted in the unbiased PCA (Supplementary Fig. 3A).

Azithromycin, clindamycin, and linezolid were chosen for further studies due to the presence of lumen-containing colonies in all three concentrations in the screen (Fig. 3F and Supplementary Fig. 3B). The three concentrations tested for each of these antibiotics indicated a potential dose-dependent response. To formally test this, a larger concentration range with 9 concentrations for the three antibiotics was then conducted in the 384-well format. All three drugs showed a dose-dependent increase in median colony area (Supplementary Fig. 3C) and percent colonies with lumens (Supplementary Fig. 3D).

We next sought to determine if the hits identified in the screen also induce re-circularization in our standard 24-well collagen cultures. SC cells were plated in type I collagen in a 24-well plate format and treated with 10, 1, or 0.1 µM of azithromycin, clindamycin, or linezolid for 14 days. The cystic conversion by the antibiotics was quantified as a circularity index, which was described previously as a relationship between colony area and perimeter, where rounder colonies had a higher circularity index[11]. Treatment with each of the antibiotics, azithromycin (Fig. 3G and Supplementary Fig. 5), clindamycin (Fig. 3H and Supplementary Fig. 5), and linezolid (Fig. 3I and Supplementary Fig. 5), resulted in a dose-dependent increase in colony circularity. Treatment with a concentration of 10 µM of each antibiotic led to a significant increase in average colony circularity compared to the untreated control. Treatment with azithromycin at 10 µM also caused a significant increase in average colony circularity compared to treatment with 0.1 µM azithromycin (Fig. 3G). We fixed and stained drug-treated colonies with the nuclear marker (DAPI), membrane/F-actin marker (Phalloidin), and an epithelial marker (E-cadherin). Compared to control colonies, colonies treated with all three antibiotics showed increased staining with E-cadherin, indicating potential emergence of an epithelial phenotype (Fig. 3J). Fixed colonies treated with azithromycin were also stained for the tight junction protein ZO-1 (Supplementary Fig. 6A). While ZO-1 was present in both untreated and treated colonies, only the antibiotic-treated colonies exhibited staining pattern consistent with its enrichment in

tight junctions (Supplementary Fig. 6B, *left panel*). These results indicate that these antibiotics reorganize the CRC 3D colonies to impart features of epithelial polarity.

For broader applicability, we tested the effects of azithromycin treatment on the morphology of an additional CRC cell line, SW480. Compared to SC cells, SW480 cells have higher baseline colony circularity, but azithromycin treatment still led to a significant increase in colony circularity (Supplementary Fig. 7A, B). A greater number of azithromycin-treated SW480 colonies had lumens and a defined edge, both indicative of restored epithelial polarity (Supplementary Fig. 7C, D). Azithromycin-treated SW480 colonies also exhibited stronger E-cadherin membrane localization at the colony periphery compared to controls (Supplementary Fig. 7E, F). Finally, unlike most data in Figs. 1–3 generated using the 384-well high-throughput format, Fig. 3G–I, and Supplementary Figs. 5–7 were produced using our standard 24-well format with two CRC lines. The consistency of our results across multiple formats supports the broader validity of our high-throughput screening format.

## Transcriptional reprogramming of CRC colonies with azithromycin, clindamycin, and linezolid treatment

To assess the impact of our hits on CRC colonies, we compared gene expression between untreated control colonies and colonies treated for 12 days with the top three antibiotics. RNA-seq PCA showed clear separation along PC1 and PC2, with azithromycin and clindamycin clustering together and linezolid positioned between them and controls (Fig. 4A). A volcano plot analysis was then conducted to highlight genes that were differentially expressed upon antibiotic treatments (Fig. 4B, and Supplementary Fig. 8A, B). CLDN2 and KIF26B were among the most upregulated, and COL9A2 and NFKB2 were among the most downregulated in azithromycin-treated colonies (Fig. 4B). Clindamycin showed a similar CLDN2/KIF26B up and COL9A2/MMP7 down pattern (Supplementary Fig. 8A), while linezolid uniquely upregulated mitochondrial genes MT-ATP8, MT-CYB, and MT-ND4L (Supplementary Fig. 8B). A heatmap of the top 50 differentially expressed transcripts for azithromycin-, clindamycin-, and linezolid-treated colonies are shared in Fig. 3D and Supplementary Fig. 9C, D respectively.

To explore pathways affected by antibiotic treatment, we further performed Gene Set Enrichment Analysis (GSEA) and WebGestalt analysis. WebGestalt analysis showed upregulation of cell-matrix adhesion, epithelial morphogenesis, and immune-related pathways in azithromycin-treated colonies (Fig. 4D). Clindamycin and linezolid both upregulated extracellular matrix organization (Supplementary Fig. 9E, F). GSEA analysis showed all three antibiotics downregulated hallmark EMT and KRAS signaling pathways (Fig. 4E, F, Supplementary Fig. 8G–J). Besides its antibiotic actions, azithromycin also displays anti-inflammatory properties, which was corroborated by our RNA-seq analysis showing downregulated hallmark inflammatory response in azithromycin-treated colonies. Azithromycin treatment also downregulated TNFα signaling, STAT signaling, and angiogenesis (Fig. 4E). Similarly, clindamycin downregulated inflammatory pathways and angiogenesis (Supplementary Fig. 8G and I), while linezolid upregulated apical junction proteins (Supplementary Figs. 8H and S8J).

Pathway analyses suggested that antibiotic treatment leads to a shift toward a more epithelial phenotype. To further characterize this, we examined expression of specific protein classes by RNA-seq. Claudins are a family of transmembrane proteins that are a critical component of tight junctions, and all three antibiotics significantly upregulated expression of claudins—CLDN2, CLDN4, and CLDN7, while azithromycin and clindamycin also upregulated CLDN3 (Supplementary Fig. 9A)[17]. This aligns with an epithelial shift, as these are key tight junction proteins and epithelial markers[17–21]. Notably, the observed CLDN7 upregulation is significant as its induced expression can restore epithelial features in CRC cells and its loss is an early event in CRC carcinogenesis[22,23]. Conversely, CLDN1 and CLDN12, which are associated with cancer invasion and progression, were significantly reduced by all three antibiotics (Supplementary Fig. 9A)[17,22,24]. Additionally, all three antibiotics significantly upregulated intermediate

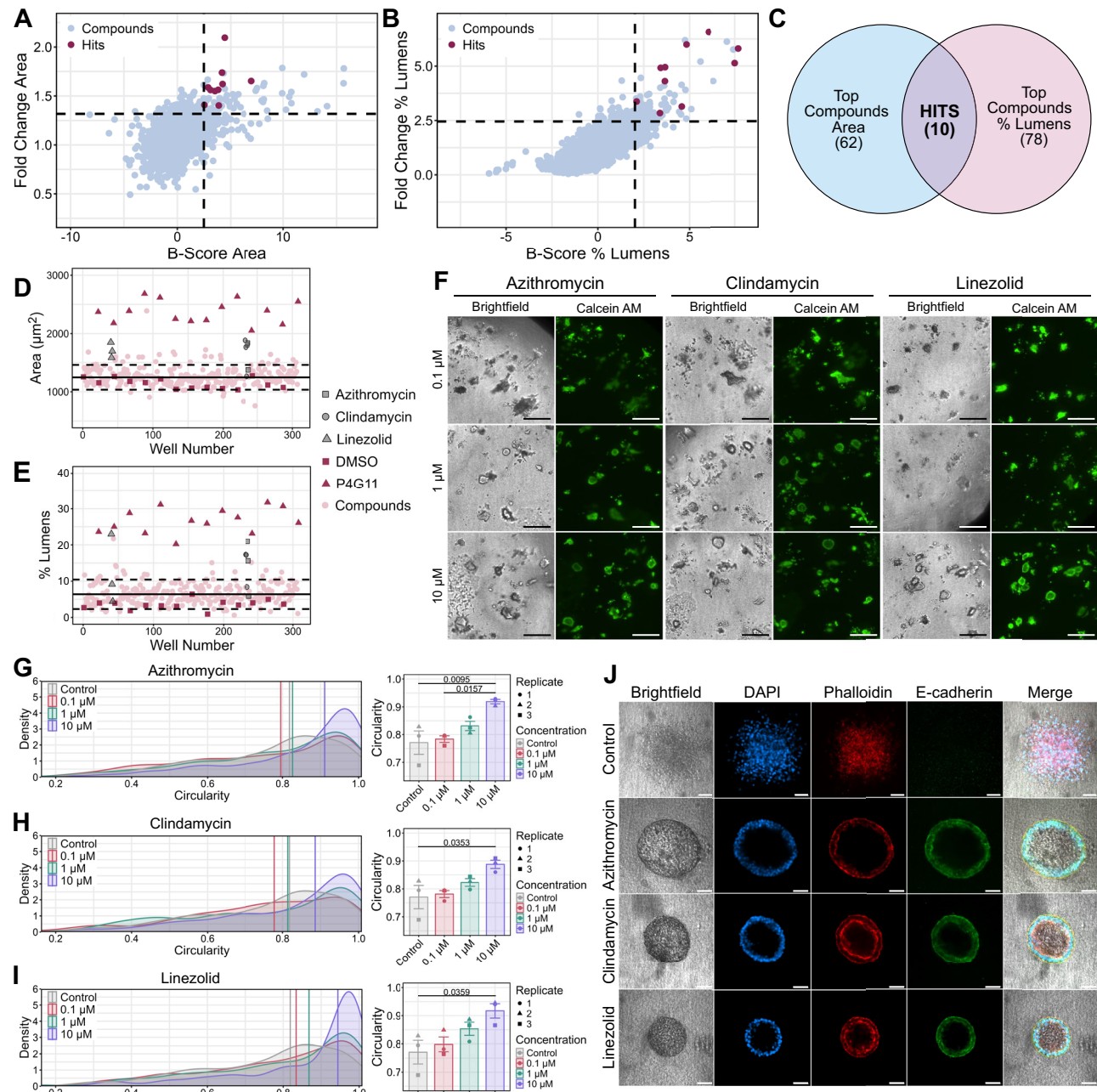

**Fig. 3 | Antibiotics identified as top three hits in screen that re-epithelialize 3D CRC colonies. A–B** Plots of median colony area (**A**) and percent colonies with lumens (**B**) for hit identification. Fold change plotted on the *Y*-axis and corresponding B-score on the *X*-axis. Dotted lines indicate the top 5% of values along its axis. **C** Venn diagram depicting selection of hits wells based on plots in (**A**, **B**). **D**, **E** Comparison of (**D**) median colony area and (**E**) percent colonies with lumens for top three hits to the plate they were located on during the screen. Well number was assigned based on order of the wells. Solid line indicates the median for all wells of the plate and the dotted lines denote the interquartile range. **F** Insets of brightfield and fluorescent (Calcein AM) confocal images of wells treated with top three hits at three concentrations from the screen. (Scale bars: 250 μm). Images taken with ImageXpress confocal HT.ai automated high-content imaging system at 4x magnification. Full well images shown in Fig. S4. **G–I** Quantification of CRC (SC cells) colony circularity when treated with (**G**) azithromycin, (**H**) clindamycin, or (**I**)

linezolid. SC cells were seeded in type I collagen and incubated with either 10 μM, 1 μM, or 0.1 μM of the indicated antibiotic for 14 days. Results are quantified as circularity index. Higher circularity indicates rounder colonies and lower circularity index indicates spiky colonies. Left panel depicts density plot of the circularity for all colonies from each treatment for all three biological replicates. Vertical line indicates the median circularity for each treatment condition. Right panel depicts median circularity for each biological replicate. $n > 150$ for each biological replicate from 3 technical replicates (wells). Plotted as mean ± SEM. One-way ANOVA with Tukey's post hoc test, *p*-values indicated. **J** Brightfield and fluorescent confocal images of SC cell colonies treated with indicated antibiotics. SC cells were seeded in type I collagen and incubated with 10 μM of the indicated antibiotic for 14 days. Collagen layer containing colonies was isolated, fixed, and stained for DAPI (blue), Phalloidin (red), and E-cadherin (green). Colonies were imaged through the equatorial plane using a 10x objective. (Scale bars: 100 μm).

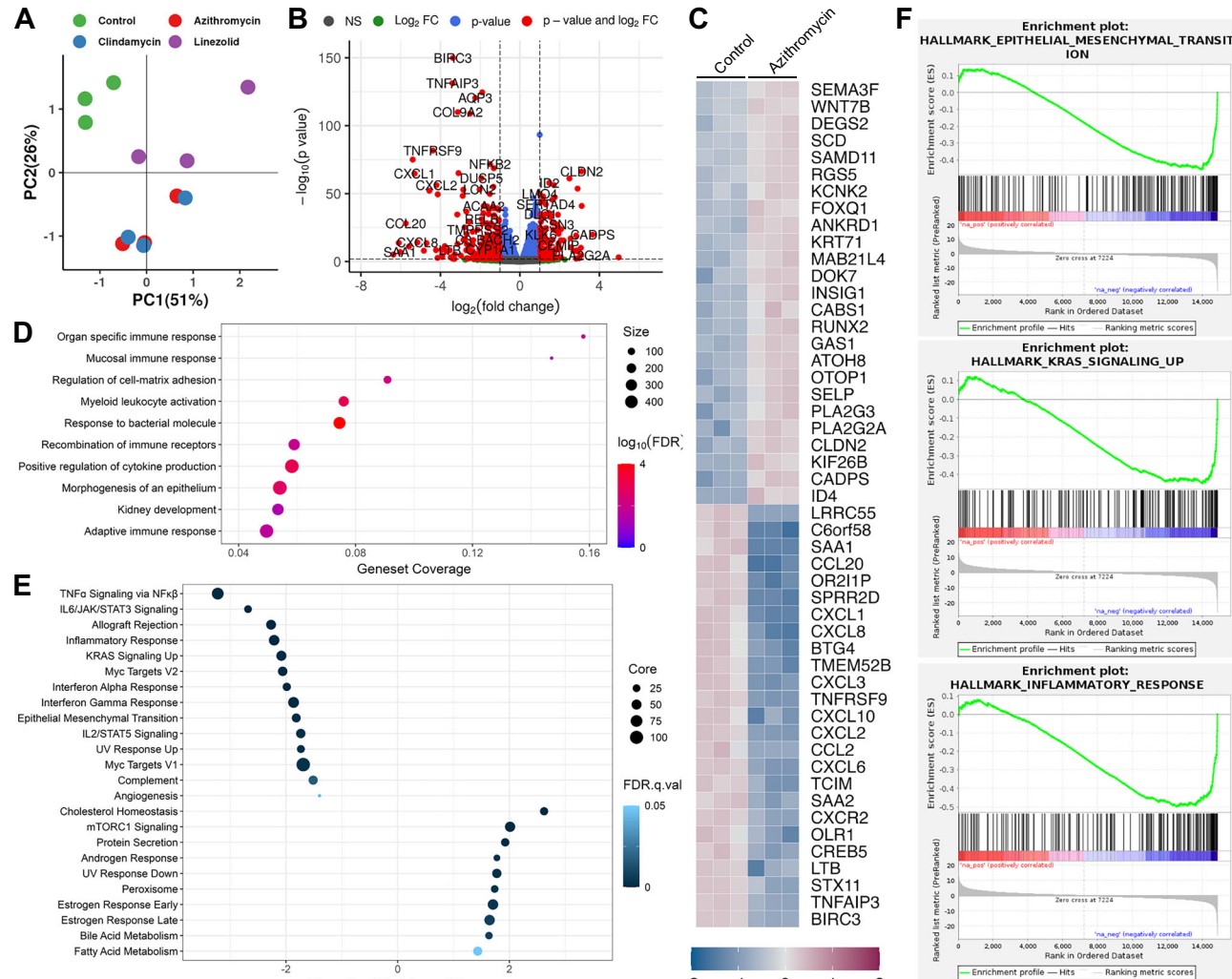

**Fig. 4 | Transcriptomic analysis of 3D CRC colonies treated with azithromycin.**
**A** Principal component analysis (PCA) plot of RNA-seq data from SC cell 3D cultures treated with azithromycin, clindamycin, and linezolid for 12 days. Each dot represents a biological replicate. **B** Volcano plot for all comparisons all correlation PCA for control and azithromycin treated 3D cultures. Differential expression analysis criteria: absolute fold change ≥ 2 and FDR adjusted *p*-value ≤ 0.05.
**C** Heatmap of top 50 differentially expressed transcripts in control vs azithromycin-treated 3D cultures in triplicate; purple = up, blue = down (expression scale in inset). **D** WebGestalt-based control vs azithromycin-treated pathway over-representation analysis. Top 10 biological processes significantly overrepresented (FDR ≤ 0.5) are

depicted with their respective enrichment ratios. **E** Hallmark gene set enrichment analysis comparison of RNA-seq data from control and azithromycin-treated 3D cultures. Hallmark pathways are indicated on the left and are represented as bubbles on the Normalized enrichment score on *x*-axis. Bubble size indicates core enrichment or number of leading-edge genes and bubble color represents FDR (false discovery rate); scales of both are indicated on the right. **F** Gene set enrichment analysis (GSEA) of RNA-seq data from control and azithromycin-treated cultures. Three select categories of interest are shown. Abbreviations: NES Normalized enrichment score, FDR False discovery rate *q*-values.

filaments (keratins)—KRT8 and KRT19, while azithromycin and clindamycin also increased KRT18 expression (Supplementary Fig. 9C). These keratins (KRT8, KRT18, and KRT19) are established epithelial markers[18–20], and their upregulation is consistent with re-epithelialization, as their loss is associated with EMT and CRC progression[25–28]. Surprisingly, only linezolid increased RNA expression of all 13 mitochondrial-encoded electron transport chain (ETC) genes, whereas azithromycin and clindamycin induced fewer ETC genes and to a lesser extent (Supplementary Fig. 9B). Taken together, these findings indicate that antibiotic treatment promotes re-epithelialization in CRC colonies, as shown by both pathway analysis and specific gene expression changes.

**Azithromycin increases the efficacy of treatment with Irinotecan**
Given the role of EMT in the development of treatment resistance by CRC, we sought to determine the effect of azithromycin on treatment response of SC cells grown in collagen. For this, we elected to focus on the interaction of azithromycin with irinotecan, which is a key component of several

chemotherapeutic regimens for the treatment of advanced CRC[29]. By itself, azithromycin had a mild effect on the number of colonies formed by SC cells in type I collagen, causing a 5% decrease in colony numbers compared to untreated cells. When cells were treated with the active metabolite of the chemotherapy irinotecan, SN-38, there was a significant drop in colony number, with an 80% decrease in number of colonies formed compared to the control (Fig. 5A, C). There was an even greater decrease in colony formation when cells were treated with both SN-38 and azithromycin compared to either drug alone. The combination therapy demonstrated almost a 95% reduction in colony formation relative to control, which was 15% greater than the observed effect when cells were only treated with SN-38 (Fig. 5A, C).

Irinotecan is a topoisomerase inhibitor that has been used to treat CRC and other cancers for more than two decades and remains a critical component of standard-of-care chemotherapy for patients with metastatic CRC[29]. Azithromycin is a broad-spectrum macrolide antibiotic, that is among the most prescribed antimicrobial drugs in the United States[30,31]. Given the

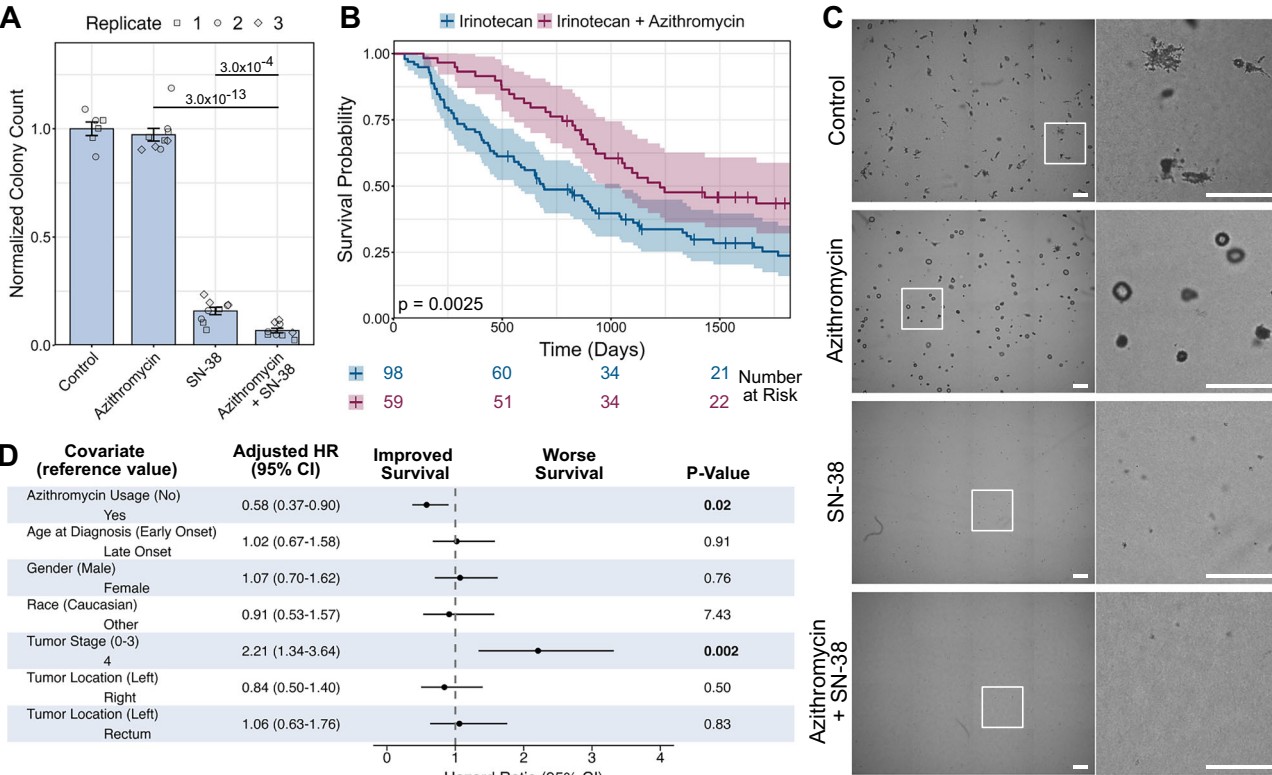

**Fig. 5 | Azithromycin enhances the efficacy of chemotherapy in CRC. A** 1250 SC cells were seeded in 3D in type I collagen and incubated for 12 days with azithromycin (5 μM), SN-38 (5 nM), or both. Points represent three technical replicates (wells) for each biological replicate. Normalized colony counts are plotted as mean ± SEM. Student unpaired *t*-test conducted for selected groups, *p*-values indicated. Berferroni's correction was used to account for family-wise error rate, with an adjusted *p*-value threshold of 0.025. **B** Kaplan-Meier plot of overall survival for patients treated with irinotecan-based chemotherapy with or without concurrent azithromycin use from a retrospective cohort of de-identified electronic health records from a single academic medical center. Manual annotation of patient's date of death was used to determine survival at 5 years (1826 days) after diagnosis. Log-rank test was used to compare treatment groups; *p*-value indicated. Risk table (bottom panel) shows patients still alive at given time for each treatment group.

**C** Representative whole well images and insets of SC cells grown in 24-well type I collagen cultures for 12 days with azithromycin (5 μM), SN-38 (5 nM), or both. Images taken using MuviCyte Live-Cell Imaging System. Boxes indicate inset location (Scale bars: 500 μm, Insets: 500 μm). **D** Forest plot showing results of multivariable cox proportional hazard model analysis. Age at diagnosis indicates age of the patient at initial diagnosis of cancer, which was broken down into early and late onset with early onset being defined as <50 years of age. Tumor stage indicates the stage of the tumor at initial diagnosis, not necessarily the stage of the cancer when irinotecan treatment was given. Patients with early-stage tumors at diagnosis were likely to have received irinotecan following disease progression or recurrence based on when irinotecan is commonly given. Manual annotation of patient's date of death was used to determine survival at 5 years (1826 days) after diagnosis.

regular use of both these drugs and the likelihood that at least some patients receiving irinotecan had also received concurrent azithromycin during their treatment course, we utilized de-identified electronic health records from an academic medical center to retrospectively interrogate cancer patient outcomes. This retrospective cohort was established with the following parameters (1) had a diagnosis of CRC, (2) was diagnosed between 1990 and 2024, (3) received irinotecan as a part of their CRC treatment and (4) stratified by whether they received a course of azithromycin while they were receiving irinotecan. Survival outcomes were determined from date of CRC diagnosis and date of death. 157 patients were included in the final cohort, 59 of whom received azithromycin at least once during their irinotecan course, and 98 who did not. A flow chart depicting the inclusion and exclusion of patients for this study is shared in Supplementary Fig. 10. At baseline there were no differences in patient age, stage at diagnosis, or primary tumor location (Supplementary Table 1). Based on univariate analysis, patients who received azithromycin had a median overall survival of 40.3 months (95% CI: 32-not reached) versus 22.4 months (95% CI: 18.9–34.2) for those who had not received azithromycin, resulting in a 5 year overall survival rate of 44% (95% CI: 32–59%) versus 24% (95% CI: 16–35%) respectively (Fig. 5B). When adjusting for covariates, compared with those who did not use azithromycin, patients who used azithromycin had a 42% lower risk of mortality (HR = 0.58, 95% CI: 0.37–0.90, *p* = 0.02) (Fig. 5D). Taken together, these data suggest a synergistic interaction between

azithromycin-induced re-epithelialization of CRC cells in our 3D collagen cultures and cell death induced by chemotherapy.

## Discussion

Treatment resistance is a major clinical concern in cancer, including CRC[32][−34]. EMT has been implicated as a resistance mechanism to chemo- and targeted therapies in both CRC and other cancers[35–37]. The changes in a cell as it goes through EMT can be observed as morphological changes in individual cells and overall colony morphology in 3D[38]. We capitalized on observable changes in colony morphology to conduct a high-throughput morphological screen to identify drugs which reverse EMT and potentially enhance chemotherapy response in CRC.

Traditionally, most high-throughput screens are conducted using cells grown in 2D, adhered to a plastic dish[12]. While a majority of 2D screens are simple cell death or viability assays, newer screens have performed microscopic analysis to extract morphological features[39]. Despite these advances, 2D cultures remain inherently ineffective in capturing cell-cell and cell-matrix interactions that are well-represented in 3D cultures. More than 90% of drugs tested in 2D ultimately fail in clinical testing, indicating a need for better in vitro 3D testing systems[40,41]. However, adoption of 3D cultures (e.g. Matrigel, collagen, hydrogel) has been slow, in part, due to technical challenges like automated handling of viscous fluids and paucity of methods to analyze the effects of drugs on 3D colonies[12]. Our high-throughput 3D drug

screen identified several distinct morphological classes of CRC colonies in collagen, revealing that specific antibiotics—azithromycin, clindamycin, and linezolid—consistently induced dramatic re-circularization and re-epithelialization of CRC colonies. This effect was robust across multiple concentrations and was validated in both high-throughput and standard 24-well formats, further validated by orthogonal RNA-seq and immuno-fluorescence assays, as well as in another CRC cell line, highlighting the reproducibility of these findings.

Our method is the first high-throughput drug screen in 3D type I collagen to be conducted in a 384-well format which utilized morphological markers to identify hits to the best of our knowledge. Not only were we able to identify drugs which induce an epithelial phenotype, which was of interest to us in this study, but an unbiased analysis of a range of morphological markers demonstrated other morphologies can be identified by this method. Limitations of the protocol were necessary to make it feasible, including a shorter incubation period compared to larger format collagen cultures and no replacement of media, which lead to a lower-than-expected percent colonies with lumens for P4G11 treated wells[14,15]. Despite these limitations, the hits identified in the screen were confirmed in the standard 24-well format collagen cultures and demonstrate a correlative relationship to improved patient response, indicating that this method has potential translational relevance. Other iterations of our 3D screen may include the addition of other ECM components, growth factors, metabolites, other cell types, and staining of colonies with other dyes as desired. This method presents a critical development in high-throughput screening providing a more physiologically relevant culturing and analysis method for future studies. Finally, given the recent US-FDA initiative to phase out animal testing in favor of human-relevant models, such as 3D organoid systems, our new 3D high-throughput morphological assay aligns directly with this shift.

The top three hits identified in our screen, azithromycin, clindamycin, and linezolid, are all antibiotics which target the bacterial ribosome to inhibit bacterial growth[42]. Previous studies of these drugs suggest off target binding to and inhibition of the mitochondrial ribosome (mitoribosome), impacting mitochondrial biogenesis, function, and mitophagy[43–49]. While normal cells may be relatively unperturbed by a modest reduction in mitochondrial function, these antibiotics would have a disproportionately larger effect on cancer cells, which have a higher metabolic burden. Mitochondrial dysfunction could also impact EMT, since it requires major metabolic changes in the cell, specifically a decrease in oxidative phosphorylation and down-regulation of mitochondrial proteins combined with increased glycolysis[50,51]. Our findings in 3D cultures suggest an impact of these antibiotics on EMT, epithelial polarity, and mitochondrial function that remained unappreciated in 2D cultures.

Our detailed RNA-seq analysis showed that all three antibiotics downregulated hallmark EMT and KRAS signaling pathways, both of which are central to cancer cell plasticity and invasiveness. Notably, azithromycin and clindamycin also suppressed inflammatory and angiogenic pathways, consistent with their known anti-inflammatory properties[30,52]. Linezolid, in contrast, uniquely upregulated mitochondrially-encoded ETC genes. At gene expression level, these antibiotics restored expression of proteins associated with tight junctions, intermediate filaments, and other epithelial markers. Together, these findings highlight common and unique antibiotic actions. The ability of antibiotics to induce re-epithelialization and suppress pro-invasive gene expression in CRC colonies suggests a previously unappreciated potential for repurposing these agents to modulate tumor cell plasticity and differentiation.

As shown in the results and discussed below, azithromycin-induced EMT reversal enhanced chemotherapy efficacy in vitro and correlated with improved chemotherapy response in CRC patients. Similarly, other combinations could be explored based on antibiotic-driven transcriptional reprogramming, such as pairing with KRAS inhibitors, given KRAS signaling suppression with all three antibiotics. A potential role of antibiotics in enhancing immunotherapy may also be considered, as azithromycin and clindamycin downregulated inflammatory and immune-evasive pathways (e.g., TNFα, STAT). These findings suggest that antibiotics may not only limit tumor cell plasticity but also modulate the tumor microenvironment in ways that could enhance immune surveillance. Our 3D culture system is well-suited for testing such combinations, with or without added cell types, or changes in composition of ECM proteins and ligands.

Azithromycin is routinely used for its antibiotic properties in the treatment of respiratory, urogenital, dermal and other bacterial infections, and for its anti-inflammatory properties in the treatment of cystic fibrosis, chronic obstructive pulmonary disease, and asthma[30]. In epithelial cells, azithromycin has been shown to maintain transepithelial resistance in the presence of virulence factors that induce permeability[53]. Azithromycin also altered processing and arrangement of tight junction adhesion molecules in airway epithelial cells[54]. These known azithromycin effects may contribute to the re-epithelialization of CRC 3D colonies. Interestingly, azithromycin is also prescribed for a non-cancerous EMT disorder of the conjunctival epithelial tissue, Pterygium[52]. Here, azithromycin is prescribed for its anti-inflammatory actions, however, our findings indicate that its anti-EMT actions might also contribute to azithromycin efficacy in this context. Cooperation of azithromycin with a chemotherapeutic is surprising but not without precedence. Azithromycin has been shown to be a potent autophagy inhibitor in cancer[55–58]. In pancreatic cancer cells, azithromycin was shown to enhance the cytotoxicity of the EGFR inhibitor gefitinib through autophagy inhibition[55]. Thus, studying the effect of azithromycin on anticancer response warrants further investigation, and our tractable 3D system presents as a robust alternative to 2D cultures.

While the widespread use of azithromycin allowed us to conduct a retrospective cohort study on its potential interaction with irinotecan, there were limitations to this study. To ensure confounding variables, such as tumor stage at diagnosis and tumor location did not significantly impact the results, we conducted a multivariate analysis. While indication for and duration of azithromycin are potentially confounding variables, these were not included in this analysis. Patients were excluded for the following factors (1) those who received azithromycin either before or after their treatment course with irinotecan-based chemotherapy (2) cancers of non-CRC origin, (3) unclear date of death. Other factors such as comorbid medical conditions and other antibiotics taken during treatment with CRC could not be taken into consideration due to small sample size and limited data availability in the electronic health record. Despite the limitations of our retrospective study, the results, and the results of our in vitro experiments, demonstrate a convincing correlation between the addition of azithromycin to irinotecan-based chemotherapy and improved 5 year survival for patients with CRC.

This 3D high-throughput drug screen in type I collagen identified compounds that enhance chemotherapy efficacy. Further validation in prospective trials and/or animal models, as well as determining the mechanism of action of these drugs are important future directions. While it is clear these drugs are inducing re-epithelialization of CRC colonies in collagen, which would account for the increased efficacy of chemotherapy treatment, the binding targets of these antibiotics which mediate this process are unknown. In addition to the drugs shared in this study, data shared for the effect of >1000 FDA-approved compounds on 3D colony morphology (see Supplementary Data 1 and associated repository[13]) may be of broader interest as an additional parameter to measure the effect of anti-cancer drugs. The results of this study demonstrate the effectiveness of 3D morphological high-throughput screens in the development of novel therapies or combination therapies for the treatment of CRC and other cancers.

## Materials and methods
### Reagents
PureCol® bovine type I collagen was purchased from Advanced Biomatrix, Inc. (San Diego, CA, USA, #5005). Unless specified otherwise, all cell culture components were purchased from Hyclone laboratories, Inc. (Omaha, NE, USA). DMEM was purchased from Corning (Corning, NY, USA, #10-027-CV). 10X MEM and GlutaMAX™ were purchased from Gibco (Waltham, MA, USA, #11430030, #35050061). Sodium hydroxide 10 Normal was purchased from VWR Chemicals BDH (Radnor, PA, USA, #BDH3247-1). Anti-Integrin β1 Antibody, clone P4G11 was purchased from Sigma-

Aldrich (Damstadt, Germany, #MAB1951). Azithromycin, clindamycin, and linezolid were purchased from Selleck Chemicals LLC (Houston, TX, USA, #S1835, #S2830, #S1408). SN-38 was purchased from MedChemExpress LLC (Princeton, NJ, USA, #HY-13704). Anti-E-cadherin was purchased from BD Transduction Laboratories (Franklin Lakes, NJ, USA, #610181). Anti-ZO-1 antibody was purchased from Invitrogen (Waltham, MA, USA, #61-7300).

384-well plates (CELLSTAR Cell Culture Microplate, Cell-Culture Treated, Flat-Bottom, with Lid) were purchased from Greiner Bio-One (Kremsmünster, Austria, #781090). Reagent reservoirs were purchased from ThermoFisher Scientific (Waltham, MA, USA, #1064-05-7). P30 Aligent Compatible tips were purchased from Fluotics (New York, NY, USA #AGI-30.ST). Calcein Green AM was purchased from Invitrogen (Waltham, MA, USA, #C1430). FDA approved drug library was purchased from Selleck Chemicals LLC (Houston, TX, USA, #L1300).

### Cell lines and cell culture (maintenance of cells on plastic)
The HCA-7 cell line was obtained from Susan Kirkland (Imperial Cancer Research Fund); generation of its derivative (SC) has been described earlier[9]. Unless otherwise indicated, cells were maintained in DMEM supplemented with 10% bovine growth serum, 1% non-essential amino acids, 1% L-glutamine, and 1% penicillin/streptomycin.

### 3D type I collagen culture in 24-well format
The concentration of collagen was 2 mg/mL (PureCol Type I Collagen, #5005) in 1X MEM (diluted from 10X MEM with sterile water) and supplemented with 10% fetal bovine serum and pH adjusted to neutral with 10 N NaOH. For a 24-well dish, three distinct collagen layers at 250 μL each were layered on top of each other (collagen layer, single-cell collagen suspension layer, and collagen layer). Cells were seeded at 5,000 cells/mL (1,250 cells/well) for colony counting and circularity experiments. After the last colony layer was polymerized, 250 μL of media with indicated treatment was added to each well. Media changes occurred every 2–3 days. Colonies were counted or imaged after 10-14 days. Colonies were counted using GelCount (Oxford Optronix) with identical acquisition and analysis settings. Alternatively colonies were assessed for circularity as described below.

### 3D type I collagen culture in 384-wells
High throughput drug screen was performed in collaboration with the Vanderbilt High Throughput Screening (VHTS) facility.

**Preparing reagents for high throughput drug screen.** The day prior to plating, 384-well plates, tips, and reagent reservoirs were placed at −20°C until used for plating. Type I collagen was prepared at a concentration of 2 mg/mL in 1X MEM (diluted from 10X MEM with sterile water), supplemented with 10% fetal bovine serum and pH adjusted to neutral with 10 N NaOH. Two collagen aliquots per round of plating were stored on ice until use. A single-cell suspension of SC cells was added to one aliquot of type I collagen at a concentration of $1.0 \times 10^5$ cells/mL (1000 cells/well) and the type I collagen was then inverted to evenly distribute cells. This aliquot was used to plate the layer of cells in collagen.

**High-throughput drug screen.** Using an Agilent BRAVO automated pipette liquid transfer system (Agilent Technologies, Santa Clara, CA, USA), chilled plates were stamped with a 5 μL bottom layer of collagen. Plates were incubated at -20°C for 15 min, with sides of the plates being tapped every 5 min to ensure the bottom of each well was entirely coated with collagen. Plates were incubated for another 15 min at 37°C. Plates were stamped with a 10 μL layer of cells in collagen using the BRAVO, then incubated at 37°C with 5% $CO_2$ for 30 min. Drugs and small molecules were transferred from the library stock plates at 10 mM DMSO solutions using an ECHO acoustic liquid transfer system (Beckman Coulter Life Science, Indianapolis, IN, USA) to 384-well drug plate. Drug plates were diluted with complete DMEM supplemented with 10% fetal bovine serum, non-essential amino acids, L-glutamine, penicillin/

streptomycin, and GlutaMAX™. 40 μL of media containing drugs and small molecules was added to the cell plates at three final concentrations (0.1 μM, 1 μM, and 10 μM). Final concentrations of drugs and small molecules account for volume of collagen in the wells. Cells were incubated for 8 days and then stained with Calcein AM at a final concentration of 3.75 μM for 3 h. Plates were imaged on an ImageXpress confocal HT.ai automated high-content imaging system (Molecular Devices, LLC, San Jose, CA, USA) using the transmitted light and FITC channels and 4x objective. Z-series were collected containing 17 planes with a step size of 50 μM and maximum intensity projections were saved.

### Analysis of 3D collagen culture in 384 wells
**PCA analysis.** Images were analyzed using the InCarta image analysis software (Molecular Devices, LLC, San Jose, CA, USA). A deep learning-based model (SINAP: Segmentation Is Not a Problem) was specifically trained to identify and segment both the cystic and spiky colonies in the transmitted light images. Details on the acquisition and definitions for each of the morphological measurements taken can be found on the Molecular Devices website (https://www.moleculardevices.com/products/cellular-imaging-systems/high-content-analysis/in-carta-image-analysis-software). Images of colonies from control treated wells were used as the training set data. Data frames were filtered to remove empty wells and incomplete data points, Scikit-learn[59] was used to generate pair-wise intervariable Pearson correlation coefficients. Using the StandardScaler class of the scikit-learn Python library, raw reads were z-score standardized per variables and principal component analysis was conducted. PC loadings were calculated: eigenvector $* \sqrt{\text{eigenvalues}}$.

**Median area, percent colonies with lumens, and total colony counts.** Fluorescent images were analyzed using the MetaXpress high content image analysis and acquisition software (Molecular Devices, LLC., San Jose, CA, USA). Intensity thresholds were used to remove out of focus colonies, monolayers, or lumens. Size filtering was used to remove colonies and lumens above or below a given size, often representing false positives. The masks generated from this pipeline was used to determine the area of individual colonies which were exported and R software (https://www.r-project.org/)[60] used to calculate the median area, number of colonies, and percent colonies with lumens for each well. Fold change median colony area and percent colonies with lumens were calculated by dividing the value for each well by the average for the DMSO control wells of the corresponding plate. B Scores were calculated, as previously described[15], for each drug treated well.

### Circularity analysis
Circularity analysis was conducted as previously described[11] but in brief; cells were seeded at 5000 cells/mL in type I collagen. Colonies were grown with indicated treatment for 10-14 days. Sixteen images per well were captured on the final day using the MuviCyte Live-Cell Imaging System (PerkinElmer). Images were analyzed in ImageJ (NIH) with the MorphoLibJ plugin[61]. Briefly, background of images was removed and a mask of colonies generated. Connected components labeling in the MorphoLibJ plugin was used to identify colonies and the size filtering used to remove colonies which were too large or small[61]. Characteristics of colonies were recorded, and colony circularity calculated using the equation: $(4*\text{Area}*\pi)/(\text{Perimeter}^2)$.

### Immunofluorescence
After 10-14 days in culture, middle layers of mature collagen cultures were separated using tweezers and washed with PBS three times, 5 min each. The collagen layer was then fixed with 4% PFA at 4°C for 1 h with gentle rocking. Collagen layer was washed again with PBS 3 times, 5 min each and then placed in IF buffer overnight at 4°C with gentle rocking (IF buffer: 1% BSA, 0.1% Triton X-100, 0.01% sodium azide in PBS). All remaining steps are performed in IF buffer at room temperature with gentle rocking. Collagen layers were then blocked with 3% normal donkey serum for 2–4 h. Primary

antibodies were added for 1 h followed by 3 × 15 min washes. Secondary antibodies were then added for 1 h followed by 3 × 15 min washes. DAPI and Phalloidin dyes were added during the secondary antibody incubation. Stained collagen layers were then placed in chamber slides with coverslip bottoms for imaging.

### Construction of patient cohort and patient data extraction

Patient data was obtained from the Synthetic Derivative, a deidentified version of Vanderbilt's electronic medical record[62–64]. Patients were initially identified based on whether they met specific screening criteria including (1) colorectal cancer diagnosis documented by ICD9 codes 153.0–153.9 or ICD10 codes C18.0-C18.9, and (2) treatment with irinotecan. This identified a total of 1381 patients including 236 patients who had received irinotecan and azithromycin, and 1145 patients who received irinotecan only. Of the patients who had received irinotecan only, 160 were randomly selected for further review and all of the 236 patients who received irinotecan and azithromycin were selected for further review. Each patient's Synthetic Derivative chart was then manually reviewed, and the following information were extracted: (1) age at diagnosis, (2) location of tumor, (3) date of diagnosis, (4) stage at diagnosis (5) date of death (if applicable), (6) date of first documented irinotecan dose, and (7) date of first documented azithromycin dose (if applicable). Patients were excluded if cancer was not colorectal in origin, irinotecan was never received, azithromycin was not given during irinotecan treatment, or date of death could not be determined. Thus, the irinotecan and azithromycin group included patients that received azithromycin during irinotecan-based chemotherapy treatment course. After manual review, 98 patients met the criteria in the irinotecan only group and 59 met the criteria in the irinotecan and azithromycin group. A schematic showing patient inclusion/exclusion decisions is additionally included in Fig. S10 flow chart.

### Statistics and reproducibility

384-well drug screen was performed at three concentrations, with one technical replicate each; 14 DMSO and P4G11 control replicates were included in each 384-well dish. 384-well dose curves were performed with three technical replicates at each concentration. All remaining experiments were performed in at least three technical and three biological replicates. All statistical analyses and graphing were performed using R software version 4.4[60]. Detailed descriptions of the statistical tools and methods for each analysis are provided in table and figure legends. For analysis of patient data, patients' characteristics were presented as medians with interquartile ranges or as frequencies with percentages. Comparisons of patient characteristics between treatment groups were made using the Wilcoxon rank-sum test or the chi-squared test. To evaluate the association between treatment and overall survival, the Kaplan-Meier method and Cox proportional hazards model were employed. For all analyses, hazard ratios (HRs), 95% confidence intervals, and $p$-values were reported.

### Reporting summary

Further information on research design is available in the Nature Portfolio Reporting Summary linked to this article.

### Data availability

Data to generate graphs and additional data from the screen available in Supplementary Data 1. Images generated in high-throughput screen are available in associated repository[13]. RNA-seq data has been deposited into the NCBI GEO under the accession number GSE304081. Any additional information related to this paper is available from the corresponding author upon reasonable request.

### Materials availability

This study did not generate unique reagents.

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

## Acknowledgements

We thank Sarah E. Glass for editing this manuscript. We thank Martha Shrubsole and Alex Hawkins for epidemiology or treatment-related clarifications. C.K.S. was supported by the Vanderbilt–Ingram Cancer Center Brock Family Fellowship. V.T.J. was supported in part by T32GM07628. K.S.L. is supported in part by funds from U01CA294527 (HTAN), R01DK103831. R.J.C. acknowledges support from NCI R35 CA197570, P50 236733, and the Nicholas Tierney GI Cancer Memorial Fund. J.A.B. is supported by an NCI R50 Research Specialist Award R50CA211206. The Vanderbilt High-Throughput Screening (VHTS) Core receives support from the Vanderbilt Institute of Chemical Biology and the Vanderbilt Ingram

Cancer Center (P30CA68485). The ImageXpress Confocal HT.ai instrument and Hamilton Verso compound storage system are housed and managed by the VHTS Core Facility, and were funded by NIH Shared Instrumentation Grant S10OD028719 and S10OD028715. B.S. is supported in part by funds from NIH/NCI R01CA248505, American Cancer Society—Research Scholar Grant RSG-20–130-01-DDC, and P50 CA236733. We acknowledge VUMC core scholarships from VUMC Cancer Center Support Grant (CCSG) P30CA068485 and VUMC Digestive Diseases Research Center grant (DDRC) P30DK058404.

## Author contributions

S.J.H., T.P.H., C.K.S., V.T.J., J.A.B., and B.S. conceived research. S.J.H., T.P.H., C.K.S., V.T.J, and C.C.W. performed experiments. S.J.H., T.P.H., S.W.K., M.A.R., Z.Z., O.K., Q.L., K.S.L., R.J.C., J.A.B., and B.S. conducted data analysis and interpretation. S.J.H. assembled figures and table. S.J.H and B.S. wrote initial draft of manuscript. All authors reviewed and approved the manuscript.

## Competing interests

The authors declare no competing interests.
