## [Transparent Peer Review file · Communications Biology]

3D collagen high-throughput screen identifies drugs that induce epithelial polarity and enhance chemotherapy response in colorectal cancer

Corresponding Author: Dr Bhuminder Singh

Version 0:

Reviewer comments:

Reviewer #1

(Remarks to the Author)

In this study, Harmych and collaborators, designed a high-throughput screen using 3D type I collagen cultures of colorectal cancer (CRC) cells to assess colony morphological changes to identified FDA-approved drugs that can to re-epithelialize colonies of this cancer type. They are reporting that of these drugs, azithromycin increased colony circularity, enhanced E-cadherin membrane localization, as well as elevated the sensitivity to irinotecan. In addition, by using a retrospective analysis of patient data showed that this antibiotic used in CRC patients treated with irinotecan enhanced the 5-year survival as compared to patients treated with chemotherapy alone. From these results, they claim that this high-throughput screen model can be useful to monitor and quantify different colony morphologies when induced by drug incubation and that these morphologic screens are important to identify novel drug candidates and synergistic mechanisms.

This is a study technically well designed, however, the conclusions they reached using the high-throughput screen in 3D type I collagen model about how the drugs effect the re-epithelization of the CRC cells, is very poor and lacks solid support. There are not mechanistic experiments, therefore it is just a preliminary study in your present form.

- In Introduction section the authors justify that EMT is a mechanism of therapy resistance, which is true, and that assessing phenotypic changes associated with this event could prove an effective method to identifying drugs that enhance current treatments. However, phenotypic changes associated with EMT is not sufficient to characterize this important cellular event. There are a wide body of evidence showing molecular and cellular aspects that characterize EMT, therefore only morphological changes are not sufficient to define it. (for instance: Nature Cancer Reviews Genetics 24, 590-609 (2023); Nature Cancer 5, 1669-1680 (2024)).

- Immunofluorescence to E-cadherin is not sufficient to characterize re-epithelization. At least other epithelial markers such as claudins and ZO proteins must be accessed by immunofluorescence and immunoblotting.

- The retrospective data of CRC patients about the possible effects of the drugs azithromycin and irinotecan must be confirmed in in vitro assays and in vivo using animal models. These experiments may confirm the mechanism of how these medications can induce the re-epithelization, which is lacking in this study.

- Finally, the authors claim in discussion section that the observable changes in colony morphology using the high-throughput model can identify drugs which reverse EMT. Again, more experiments must be carried out to confirm this, including the analyses of epithelial and mesenchymal markers as well as changes in the migration and invasion potential.

I'm sorry I cannot be more positive, but I feel that in addition to improving the technical quality of the paper, authors should actively seek for novel findings that move this field forward.

Reviewer #2

(Remarks to the Author)

The manuscript is focused on designed high-throughput screen using 3D type I collagen cultures of human colorectal cancer (CRC) cells to assess morphological changes in colonies and identified several FDA-approved drugs that re-epithelialize CRC colonies.

This approach offers a new potential therapeutic strategy to improve CRC treatment outcomes and demonstrate the effectiveness of 3D phenotypic high-throughput screens in the development of novel therapies for CRC

patients.

Overall, this is an interesting and comprehensive study, and deserves a publication in Communications Biology following addressing some comments:

- 1) Please indicate the exact p-value for each experiment in figure legend.
- 2) For Figure 4A, it may be best to show also images of colonies.
- 3) Figure 4 B. Please indicate the test used to assess statistical significance of Kaplan Meier curves (log-Rank test?).
- 4) Figure S4D. Please indicate upper the graphs the antibiotics as in fig. S5A and B
- 5) In Mat&Met you indicate that "92 patients met the criteria in the irinotecan group" but Table S1 reports 94 patients.
- 6) Please reports magnification for each IF images.
- 7) The manuscript needs revision of the grammar.
- 8) In a few places in the Manuscript there are write errors (eg figure legend, Mat&met section immunofluorescence and discussion).

Reviewer #3

(Remarks to the Author)

In this manuscript, Harmych and colleagues present a new 3D type I collagen screen, which can be conducted in 384-well plates, facilitating high-throughput screenings. As epithelial-to-mesenchymal transition (EMT) leads to therapy resistance, the Authors employed the method developed to identify drugs causing colonies re-epithelialization. The main compound identified in the screening was the antibiotic azithromycin, which triggered a morphological change from spiky to more circular colonies. Moreover, this drug increased the sensitivity to irinotecan in vitro, as well as improving the survival of colorectal cancer patients who received irinotecan.

Despite the evidences described in this manuscript are very interesting and promising, there are some aspects that the Authors should address in a more robust manner.

Major comments

1. Figure 1D. The Authors indicate the presence of 5 different morphological clusters, in which the colony morphology changed following the drug treatment. However, the colony morphology of the control wells is not shown, so it is impossible to compare the changes in morphology.
2. Figure 2B. Looking at the heatmap, it seems that characteristics like "minor axes", "entropy", "energy" or "perimeter" had a higher correlation to PC1. However, the characteristics "Percent colonies with lumens" and "median colony area" were selected for further studies. These characteristics may be easier to assess experimentally, but what is the reason behind this decision?
3. Patient's cohort. Patients who received Azithromycin 3 months before or during the irinotecan treatment were selected for the study. How was this timing chosen? How long do the azithromycin effects last? Will you find any differences in survival if patients are stratified depending on azithromycin before or during irinotecan treatment?

Minor comments

1. Several times along the manuscript, the word "morphologic" is written instead of "morphological".
2. Abstract: "in part, due the complete failure of typical live/dead 2D high-throughput screens to capture morphology or lack of robustness of 3D screens". Should be due to the complete...
3. Abstract: "in part, due the complete failure of typical live/dead 2D high-throughput screens to capture morphology or lack of robustness of 3D screens". Should be or the lack of...
4. Abstract: "Here, we designed high-throughput screen using 3D type I collagen cultures of CRC cells to assess morphological changes in colonies and identified several FDA-approved drugs that re-epithelialize CRC colonies". Should be we designed a high-throughput...
5. Results: "Using these parameters... with the drugs in the screen". Which parameters, all the ones analyzed or the ones that contributes the most to PC1 and PC2?
6. Results: "While the InCarta software from Molecular Devices... an accurate border for most colonies". Reformulate the sentence.
7. Results: "To overcome this limitation, we optimized and employed an alternate method... were effectively generated using Calcein AM staining and the MetaExpress image analysis software". Alternate should be changed to alternative, alternate means "to happen or exist one after the other repeatedly".
8. Results: "The parameters, colony size and % colonies with lumens... our drug screen that induce epithelial polarity". Be consistent and write percent instead of %, as in the rest of the manuscript.

9. Results: "Fold-change and B-scores were used to plot each well and then the top 5% for each calculated (Figures 3A and 3B)". For each was calculated.

10. Results: "To formally test this, a larger concentration range with 10 concentrations for the three antibiotics was then conducted in the 384-well format". Only 9 concentrations are represented, as written in Figure S4C-D legend.

11. Results: "Based on univariate analysis, patients who received azithromycin had a 47% better rate of survival at the 5-year mark compared to patients who only received irinotecan (Figures 4B and S6)". This exactly corresponds to Figure S6A.

12. Discussion: "Also, data shared for the effect of more than 1000 FDA-approved compounds on 3D colony morphology (see Data Table 1 in supplemental materials) may be of broader interest as another parameter to measure the effect of anti-cancer drugs and potential combinations". I think the Authors should indicate the presence of all the data in the main text and not only in the discussion.

13. Methods: "Anti-E-cadherin (BD Transduction Laboratories #610181 mouse antibody)". The sentence is incomplete.

14. Methods: Non-essential amino acids, L-glutamine, and penicillin/streptomycin are used at 1X? Please indicate it.

15. Methods: "For a 24-well dish, three distinct collagen layers at 250 μ L each were layered in top of each other (collagen layer, single-cell collagen suspension layer, and collagen layer)". On top of each other.

16. Methods: "Z-series were collected containing 17planes with a step size of 50 μ M and max projections were saved". Maximal instead of max.

17. Methods: "Patients who met screening criteria were initially identified based on whether they metspecific screening criteria within the last 5 years including 1) colorectal cancer diagnosis documented by ICD9codes 153.0-153.9 or ICD10 codes C18.0-C18.9, and 2) treatment with irinotecan" There is a repetition of "met screening criteria" that render the sentence difficult to understand.

18. Methods: "several DMSOand P4G11 control replicates were included in each 384-well dish". How many DMSO and how many P4G11 controls were included in each plate?

19. Figure 1A: "Next, 1,059 compounds from an FDA-approved drug library were added at threeconcentrations and incubated for 8 days". From a FDA-approved drug library

20. Figure 3D-E: "Comparison of (D) median colony area and (E) percent colonies with lumens for top three hits to plate theywere on during the screen". Consider reformulating the last part of the sentence, for example for top three hits based on the plate they were located during the screening

21. Figure 3F and S4B: Figure 3F showed the magnification of the 10uM concentration. Could the Authors also include the magnification for the other concentrations to properly see the lumens?

22. Figure 3F: In the case of Azithromycin the lumens are not very clear. Consider using another representative image.

23. Figure 3F: Indicate that the representative images correspond to the 10uM concentration observed in Figure S4B.

24. Figure 3G: What is written in the main text does not correspond with the significance marked in the Azithromycin graph.

25. Figure 3G: please indicate the number of colonies analyzed in each biological replicate.

26. Figure 4A: "1,250 cells SC cells were seeded in 3D in type I collagen and incubated for 14 days with azithromycin (10 μ M), SN-38 (20 nM), or both". The word cells is repeated twice.

27. Figure S2 and Figure S4B: I would include these two figures in the corresponding main figures to have a proper view of the well and the magnified part.

28. Figure S4C-D: which graph corresponds to each antibiotic? Please indicate it

29. Figure S5A-B: "(A-B) Comparison of (A) median colony area and (B) percent colonies for additional hits to the plate they werelocated on for the screen". Consider reformulating the last part of the sentence.

30. Figure S5C: With that magnification the lumens are not visible

31. Figure S6B-C: The Authors do not comment anything about them in the main text.

Version 1:

Reviewer comments:

Reviewer #2

(Remarks to the Author)

The authors have responded to all my comments. The new version of the manuscript are more detailed and I agree for the publication in Communications Biology.

Reviewer #3

(Remarks to the Author)

The authors satisfactorily addressed all the issues I raised.

Bhuminder Singh, PhD
Assistant Professor of Medicine and Cell and Development Biology
Division of Gastroenterology, Hepatology, and Nutrition

Georgios Giamas, PhD
Editorial Board Member, Communications Biology

May 22, 2025

Dear Dr. Giamas,

Thank you for giving us the opportunity to respond to reviewers' insightful comments on our manuscript "**3D collagen high-throughput screen identifies drugs that induce epithelial polarity and enhance chemotherapy response in colorectal cancer**". In the enclosed revised manuscript, we have diligently addressed all reviewer concerns and incorporated their valuable suggestions. Our revision efforts have now been included in a full new main figure, five supplemental figures, and updates and panel additions to majority of earlier main and supplemental figures and associated text (see changes table below). We are confident these changes have substantially improved the manuscript and hope it is now suitable for publication in *Communications Biology*.

We were encouraged by the largely positive reception, with Reviewer 2 remarking that it "*deserves a publication in Communications Biology*" and Reviewer 3 finding the work "*very interesting and promising*." Their minor suggestions have been incorporated. Per Editor's suggestion we have also included experiments with another CRC cell line and further performed extensive validation with additional assays (RNA-seq, microscopy, immunofluorescence).

Reviewer 1 had major concerns regarding a deeper analysis of EMT reversion and additional *in vivo* characterization. While we agree that *in vivo* analysis is an important next step for mechanistic understanding that we intend to pursue, we believe it extends beyond the scope of the current manuscript. However, to address the core of these concerns, we conducted a detailed transcriptomic analysis of our cells cultured in 3D in the presence of three lead candidates. Significantly, this analysis demonstrated a downregulation of EMT signatures for all three compounds. These new findings, which strongly support our primary conclusions, are now detailed in a new main figure and two accompanying supplementary figures, outlining RNA-seq data. In another orthogonal assay, we also showed better organization of the tight junction protein ZO-1 in 3D colonies upon antibiotic treatment, indicating re-establishment of epithelial phenotype.

Reviewer 1 also questioned the novelty and impact of our study, citing lack of *in vivo* data. However, we submit that implementation of a 3D HTS assay in type I collagen is unprecedented and marks a significant development in the field. Using morphological measurements, rather than standard single-readout assays, is also a new concept in drug screening. Combined, 3D HTS assays with morphological readouts present a robust opportunity for greater clinical relevance. Furthermore, we provide morphological data for colonies treated with over 1,000 FDA-approved drugs, along with colony/well images, which is a valuable resource for other researchers in their own pursuits. Reviewer 3 also encouraged us to highlight this potential contribution, and we are

preparing to make this dataset publicly available following publication, please see associated figshare submission. Also, as members of the NCI Human Tumor Atlas Network (HTAN) investigators, we would like this manuscript to be considered for inclusion in the Nature HTAN collection. Finally, given the recent US-FDA initiative to phase out animal testing in favor of human-relevant models, such as 3D organoid systems, our new 3D HTS morphological assay aligns directly with this shift.

Below, we address each of the reviewers' major and minor concerns in a point-by-point manner; accompanying major changes in the revised manuscript are highlighted in red. See below table which outlines changes made to each figure.

Figure	Revisions
1	 - Panel B added - Panel E added - Control wells added to panels E and F
2	 - P-values added to panels D and E
3	 - Insets from additional concentrations added to panel F - P-values added to panels G-I
4	 - New figure (A-F)
5	 - Previously Figure 4 - Experiments for panel A redone with lower azithromycin and SN-38 concentrations - P-values added to panel A - Panel C added - Patient cohorts re-established and analysis reconducted for panels B and D - Risk table added to panel B
S1	 - No changes made
S2	 - No changes made
S3	 - Previously Figure S4 - Additional insets indicated in panel B - Labels added indicating antibiotics used to generate graphs in panels C and D
S4	 - Previously Figure S5 - Panel D added
S5	 - New figure
S6	 - New figure (A-B)
S7	 - New figure (A-F)
S8	 - New figure (A-J)
S9	 - New figure (A-C)
S10	 - New figure
Table S1	 - Patient cohorts re-established and analysis re-conducted
Table S2	 - No changes made

Reviewer 1

- I. *This is a study technically well designed, however, the conclusions they reached using the high-throughput screen in 3D type I collagen model about how the drugs effect the re-epithelization of the CRC cells, is very poor and lacks solid support. There are not mechanistic experiments, therefore it is just a preliminary study in your present form.*

Response: Thank you for recognizing the sound technical design of our study. We also feel the design and implementation of a 3D high-throughput drug screen in type I collagen represents a critical step forward in this field. 3D cultures have been shown to be much more clinically relevant than 2D screens, as they replicate more of the cell-cell and cell-matrix interactions that occur in the body. Ours was first among many attempts to successfully conduct a high-throughput drug screen in collagen. We took the assay one step further by employing a morphological read-out rather than a standard single read-out assay (e.g., cell viability), which only enhances the information that can be obtained from the screen. Follow up has demonstrated the reproducibility of the results from the screen and additional transcriptomic analysis has demonstrated a decreased EMT signature upon treatment with three top candidates (new Fig. 4 and Figs. S8, S9). These data provide initial insights into potential mechanisms (e.g. the potential mitochondrial impact of Linezolid; new Figs. S8B, S9B) that we intend to pursue. However, the significant impact of this manuscript remains the technical advance of establishing a robust method of conducting a 3D HTS morphological screen. (Lines 201-239, 318-335).

- II. *In Introduction section the authors justify that EMT is a mechanism of therapy resistance, which is true, and that assessing phenotypic changes associated with this event could prove an effective method to identifying drugs that enhance current treatments. However, phenotypic changes associated with EMT is not sufficient to characterize this important cellular event. There are a wide body of evidence showing molecular and cellular aspects that characterize EMT, therefore only morphological changes are not sufficient to define it. (for instance: Nature Cancer Reviews Genetics 24, 590-609 (2023); Nature Cancer 5, 1669-1680 (2024)).*

Response: We agree that morphological conversion alone is not sufficient to define EMT reversal. To accompany morphological change, we previously showed elevated E-cadherin levels after antibiotic addition (immunofluorescence, Fig. 3J). In this revision, we have additionally demonstrated enrichment of ZO-1 to tight junctions after azithromycin treatment (immunofluorescence, new Fig. S6) and additional brightfield images of colonies treated with all three antibiotics to consider further (new Fig. S5). (Lines 186-190).

Most importantly, we conducted a detailed RNA-seq analysis of colonies treated with azithromycin, clindamycin, and linezolid for 12 days in 3D collagen cultures (Figs. 4, S8, and S9). These results show a significant reduction in hallmark EMT signature following treatment with all three antibiotics (Figs. 5, S8). Additionally, RNA-seq analysis revealed increased expression of claudins (claudins 2, 4, 7) and keratins (keratin 8, 18, 19), indicating a shift toward a more epithelial phenotype upon antibiotic treatment (new Fig. S9). Importantly, claudin 7 expression has previously been shown to induce mesenchymal-to-epithelial transition (MET) to inhibit CRC tumorigenesis (PMID: 25500541). With this additional experimental evidence, we are confident that these antibiotics induce an epithelial phenotype. Additional explanations are provided on page 1 of this letter and in response to other queries below. (Lines 201-239, 318-335)

- III. *Immunofluorescence to E-cadherin is not sufficient to characterize re-epithelization. At least other epithelial markers such as claudins and ZO proteins must be accessed by immunofluorescence and immunoblotting.*

Response: Following this suggestion, we performed additional immunofluorescence of azithromycin-treated colonies using the tight-junction protein, ZO-1. While ZO-1 was present in both untreated and treated colonies, only the antibiotic-treated colonies exhibited staining pattern consistent with its enrichment in tight junctions (Fig. S6). Based on reviewer suggestion, we next probed other key class of tight junction proteins, claudins, by RNA-seq analysis. Here, claudin 2, 4, and 7 were significantly enriched upon treatment with all three antibiotics (new Fig. S9A). Moreover, epithelial keratins 8, 18, and 19 were also significantly enriched (RNA-seq analysis) upon antibiotic treatment (new Fig. S9C). Additionally, GSEA analysis of RNA-seq data from antibiotic-treated colonies revealed a significant reduction in the hallmark EMT signature, supporting the transition to a more epithelial phenotype in CRC cell colonies treated with the identified hits. (lines 186-190, 201-239, 318-335)

- IV. *The retrospective data of CRC patients about the possible effects of the drugs azithromycin and irinotecan must be confirmed in in vitro assays and in vivo using animal models. These experiments may confirm the mechanism of how these medications can induce the re-epithelialization, which is lacking in this study.*

Response: We recognize the inherent limitations of a retrospective patient study in evaluating azithromycin's impact on irinotecan efficacy *in vivo*. To mitigate this, we did perform *in vitro* 3D colony growth (updated Fig. 5B) and morphology (new Fig. 5C) assays for the irinotecan and azithromycin combinations that support irinotecan-azithromycin cooperation. As noted on page 1, further *in vivo* and *in vitro* studies may extend beyond the scope of this report but are planned for future research. Investigating the mechanism of antibiotic-induced re-epithelialization would require extensive analysis, including identifying binding targets, signaling pathways, and other contributing factors. We emphasize that primary objective of this study was to validate the effectiveness of a novel 3D high-throughput screening platform in identifying drugs with potential clinical relevance, which we believe we have successfully demonstrated.

- V. *Finally, the authors claim in discussion section that the observable changes in colony morphology using the high-throughput model can identify drugs which reverse EMT. Again, more experiments must be carried out to confirm this, including the analyses of epithelial and mesenchymal markers as well as changes in the migration and invasion potential.*

Response: This has largely been addressed above. To reiterate, extensive validation with additional assays like RNA-seq (EMT signature down, polarity proteins claudins and keratins up), microscopy (decreased ECM projections, emergence of hollow lumens), and immunofluorescence (tight junction relocalization of ZO-1) during this revision confirms EMT reversal following antibiotic treatment (Figs. 3, 4, 5C, S3, S5, S6, S7, S8, S9). The loss of a spiky phenotype, or the reduction in protrusions into the matrix, which is also quantified by the circularity index (Fig. 3G-I) demonstrates reduced invasion into the matrix (additional microscopic examples in Figs. 3F/J, 5C, S4D, S5, S6, and S7).

- VI. *I'm sorry I cannot be more positive, but I feel that in addition to improving the technical quality of the paper, authors should actively seek for novel findings that move this field forward.*

Response: We sincerely appreciate your time in reading and providing feedback. To emphasize our key contributions, we present the first use of a 3D high-throughput drug screen in type I collagen, along with one of the earliest examples of a screen using phenotypic measurements to assess drug effectiveness – both of which represent substantial advances in high-throughput screening. Additionally, we offer a repository of images that can serve as

a valuable resource for researchers evaluating the impact of their drug of interest on CRC colony morphology. Moving forward, mechanistic studies will be conducted for the identified hits. We hope that this work is recognized as a significant technical advance, featuring extensive validation of EMT phenotype through orthogonal approaches, and is now deemed suitable for publication in *Communications Biology*.

Reviewer 2

- I. Overall, this is an interesting and comprehensive study, and deserves a publication in *Communications Biology* following addressing some comments:

Response: We are delighted that you deemed our manuscript deserving of publication in *Communications Biology*. We really appreciate you taking the time to read and provide feedback on our manuscript.

- II. Please indicate the exact *p*-value for each experiment in figure legend.

Response: *p*-values have now been added to figures and figure legends. (Figs. 2D, 2E, 3G-I, 5A, S7B, S9)

- III. For Figure 4A, it may be best to show also images of colonies.

Response: We have now included representative full well and inset images of colonies for Figure 4A (now Fig. 5A). They are included in Figure 5C.

- IV. Figure 4 B. Please indicate the test used to assess statistical significance of Kaplan Meier curves (log-Rank test?).

Response: The Log-rank test was used to determine this value, which has now been explicitly noted in the figure legend for Figure 4B (now Fig. 5B). Additionally, this figure has been updated to reflect a broader and more stringent patient cohort for co-treatment, incorporating patients beyond the last five years and refining the timing of azithromycin administration to align strictly with irinotecan-based treatment. (Lines 256-273, 355-361, 476,488).

- V. Figure S4D. Please indicate upper the graphs the antibiotics as in fig. S5A and B

Response: Labels have now been added to indicate the antibiotics (now Fig. S3C/D).

- VI. In Mat&Met you indicate that "92 patients met the criteria in the irinotecan group" but Table S1 reports 94 patients.

Response: Patient cohorts have now been revised based on the updated analysis, which now includes 98 patients in the irinotecan-only group and 59 in the azithromycin+irinotecan group. We have thoroughly verified that these numbers are now accurately reported in both the results and methods sections. (Table S1, Lines 262-264, 476-279).

- VII. Please reports magnification for each IF images.

Response: Magnification has been added to all figure legends containing images. (Figs. 1E-F, 2F, 3F, 3J, 5C, S3B, S4C-D, S5, S6, S7C-F)

- VIII. The manuscript needs revision of the grammar.

Response: Grammar in the entire manuscript has been reviewed.

- IX. In a few places in the Manuscript there are write errors (e.g., figure legend, Mat&met section immunofluorescence and discussion).

Response: Written errors have been addressed throughout the manuscript.

Reviewer 3

- I. *Despite the evidence described in this manuscript are very interesting and promising, there are some aspects that the Authors should address in a more robust manner.*

Response: We appreciate the time you took to read and provide constructive feedback on our manuscript. We specifically appreciate you calling our study very interesting and promising.

Major comments

- I. *Figure 1D. The Authors indicate the presence of 5 different morphological clusters, in which the colony morphology changed following the drug treatment. However, the colony morphology of the control wells is not shown, so it is impossible to compare the changes in morphology.*

Response: Apologies for this omission, images for the control wells have now been added to Figure 1 for comparison.

- II. *Figure 2B. Looking at the heatmap, it seems that characteristics like “minor axes”, “entropy”, “energy” or “perimeter” had a higher correlation to PC1. However, the characteristics “Percent colonies with lumens” and “median colony area” were selected for further studies. These characteristics may be easier to assess experimentally, but what is the reason behind this decision?*

Response: We acknowledge that other indicated characteristics exhibited a stronger correlation for PC1 in Fig. 2B, underscoring the unbiased, unsupervised strength of automated quantification. While these parameters may prove valuable following further validation, particularly for assessing unknown drug effects, our primary focus in tracking EMT reversal was to ensure visual confirmation of morphological conversion, minimizing false positives. Consequently, “Median colony area” and “percent colonies with lumens” were the most intuitive and functionally relevant choices, since they correspond to the cystic morphology we aimed to capture. While other characteristics contributed more significantly, they were not easily detectable by the human eye. There, we prioritized these parameters for their visual clarity and biological significance. This rationale has been clarified in the text. (Lines 124-134).

- III. *Patient’s cohort. Patients who received Azithromycin 3 months before or during the irinotecan treatment were selected for the study. How was this timing chosen? How long do the azithromycin effects last? Will you find any differences in survival if patients are stratified depending on azithromycin before or during irinotecan treatment?*

Response: We apologize for this confusion, previous timing of azithromycin administration relative to irinotecan treatment was imprecise, as it broadly included all patients who had potentially received both treatments within the last five years in the BioVU Synthetic Derivative, without confirming exact timing. Since our intended comparison was between irinotecan-only and irinotecan+azithromycin groups, we refined the irinotecan+azithromycin cohort to include only those who received azithromycin specifically during irinotecan treatment. To maintain a sufficiently large cohort for analysis, we removed the 5-year restriction and instead considered all relevant patients within the Synthetic Derivative. The irinotecan-only cohort was also redefined using this expanded timeframe. While the duration of azithromycin’s effect remains uncertain, restricting its timing to the course of irinotecan treatment ensures any potential impact on irinotecan efficacy is accurately captured. The

revised manuscript has been updated to incorporate these changes (e.g., new Fig. 5B/D). (Lines 256-273, 476-288).

To specifically evaluate the duration of azithromycin's effects on irinotecan efficacy, an additional comparison may be necessary, as suggested: "examining survival differences when patients are stratified based on azithromycin administration before or during irinotecan treatment." However, we plan to conduct this analysis in future studies once a larger patient dataset becomes available, as the current dataset is not sufficiently large for this assessment. Alternatively, xenograft models could provide a more feasible approach for directly testing this in a controlled experimental setting.

Minor comments

- I. *Several times along the manuscript, the word "morphologic" is written instead of "morphological".*
Response: We have now changed the word morphologic to morphological throughout the manuscript.
- II. *Abstract: "in part, due the complete failure of typical live/dead 2D high-throughput screens to capture morphology or lack of robustness of 3D screens". Should be due to the complete...*
Response: "In part," has been removed from this sentence as suggested. (Line 31).
- III. *Abstract: "in part, due the complete failure of typical live/dead 2D high-throughput screens to capture morphology or lack of robustness of 3D screens". Should be or the lack of...*
Response: "The" has been added to this sentence as suggested. (Line 32).
- IV. *Abstract: "Here, we designed high-throughput screen using 3D type I collagen cultures of CRC cells to assess morphological changes in colonies and identified several FDA-approved drugs that re-epithelialize CRC colonies". Should be we designed a high-throughput...*
Response: "Here," has been removed from the sentence for clarity. (Line 33).
- V. *Results: "Using these parameters... with the drugs in the screen". Which parameters, all the ones analyzed or the ones that contributes the most to PC1 and PC2?*
Response: This section has been updated for clarity. (Lines 92-100).
- VI. *Results: "While the InCarta software from Molecular Devices... an accurate border for most colonies". Reformulate the sentence.*
Response: This section has been updated for clarity. (Lines 139-141).
- VII. *Results: "To overcome this limitation, we optimized and employed an alternate method... were effectively generated using Calcein AM staining and the MetaExpress image analysis software". Alternate should be changed to alternative, alternate means "to happen or exist one after the other repeatedly".*
Response: This section has been updated for clarity. (Lines 139-141).
- VIII. *Results: "The parameters, colony size and % colonies with lumens... our drug screen that induce epithelial polarity". Be consistent and write percent instead of %, as in the rest of the manuscript.*
Response: % has been changed to the written-out "percent" in this location and throughout the manuscript as needed.

- IX. *Results: "Fold-change and B-scores were used to plot each well and then the top 5% for each calculated (Figures 3A and 3B)". For each was calculated.*
Response: Sentence has been changed as suggested. (Lines 151-153).
- X. *Results: "To formally test this, a larger concentration range with 10 concentrations for the three antibiotics was then conducted in the 384-well format". Only 9 concentrations are represented, as written in Figure S4C-D legend.*
Response: This is correct, only 9 drug concentrations were tested, and the text has been changed. (Line 171).
- XI. *Results: "Based on univariate analysis, patients who received azithromycin had a 47% better rate of survival at the 5-year mark compared to patients who only received irinotecan (Figures 4B and S6)". This exactly corresponds to Figure S6A.*
Response: Figure 6A has been removed since the data was repetitive. In-text callouts have been removed as well.
- XII. *Discussion: "Also, data shared for the effect of more than 1000 FDA-approved compounds on 3D colony morphology (see Data Table 1 in supplemental materials) may be of broader interest as another parameter to measure the effect of anti-cancer drugs and potential combinations". I think the Authors should indicate the presence of all the data in the main text and not only in the discussion.*
Response: A sentence has been added to the abstract and results section to indicate the presence of the supplemental excel file with the measurements and the repository with images from the screen. (Lines 35, 91-92, 368).
- XIII. *Methods: "Anti-E-cadherin (BD Transduction Laboratories #610181 mouse antibody)". The sentence is incomplete.*
Response: This sentence has been updated as suggested. (Lines 381-382).
- XIV. *Methods: Non-essential amino acids, L-glutamine, and penicillin/streptomycin are used at 1X? Please indicate it.*
Response: Amount of these reagents in the cell culture media have been added to the methods. (Lines 394-395).
- XV. *Methods: "For a 24-well dish, three distinct collagen layers at 250 μ L each were layered in top of each other (collagen layer, single-cell collagen suspension layer, and collagen layer)". On top of each other.*
Response: This sentence has been changed as suggested. (Line 400).
- XVI. *Methods: "Z-series were collected containing 17planes with a step size of 50 μ M and max projections were saved". Maximal instead of max.*
Response: This sentence has been changed as suggested. (Line 431).
- XVII. *Methods: "Patients who met screening criteria were initially identified based on whether they met specific screening criteria within the last 5 years including 1) colorectal cancer diagnosis documented by ICD9codes 153.0-153.9 or ICD10 codes C18.0-C18.9, and 2) treatment with irinotecan." There is a repetition of "met screening criteria" that render the sentence difficult to understand.*
Response: Entire methods section outlining the selection of patients for the retrospective study has been rewritten for clarity. Supplemental Figure S10, a flow chart outlining the

selection of patients for the study, has also been added to assist readers in understanding this process. (Lines 476-488).

XVIII. *Methods: "several DMSO and P4G11 control replicates were included in each 384-well dish". How many DMSO and how many P4G11 controls were included in each plate?*

Response: 14 wells each of DMSO and P4G11 per plate (1 column each, excluding edge wells). This information has been added to the text. (Lines 491-492).

XIX. *Figure 1A: "Next, 1,059 compounds from an FDA-approved drug library were added at three concentrations and incubated for 8 days". From a FDA-approved drug library*

Response: After reevaluating this, we confirmed that "an FDA" is the correct usage. In the case of acronyms, "an" is appropriate when its pronunciation begins with a vowel sound, and FDA starts with a vowel sound ("eff"). (Line 538).

XX. *Figure 3D-E: "Comparison of (D) median colony area and (E) percent colonies with lumens for top three hits to plate they were on during the screen". Consider reformulating the last part of the sentence, for example for top three hits based on the plate they were located during the screening*

Response: This sentence has been rewritten for clarity. (Lines 577-579).

XXI. *Figure 3F and S4B: Figure 3F showed the magnification of the 10uM concentration. Could the Authors also include the magnification for the other concentrations to properly see the lumens?*

Response: Insets of the additional concentrations have now been added for each of the hits (Fig. 3F).

XXII. *Figure 3F: In the case of Azithromycin the lumens are not very clear. Consider using another representative image.*

Response: As suggested, an alternative representative inset has now been added to better demonstrate the effect of azithromycin (Fig. 3F).

XXIII. *Figure 3F: Indicate that the representative images correspond to the 10uM concentration observed in Figure S4B.*

Response: Concentration labels are added in the updated Fig. 3F, which now also includes all three concentrations (0.1, 1, and 10 μ M).

XXIV. *Figure 3G: What is written in the main text does not correspond with the significance marked in the Azithromycin graph.*

Response: Thank you for noticing this. Significance bars were incorrectly drawn in and have now been adjusted so they are accurate.

XXV. *Figure 3G: please indicate the number of colonies analyzed in each biological replicate.*

Response: The number of colonies varies across biological replicates, but at least 150 colonies for each biological replicate were quantified in this assay. This has now been noted in the figure legend. (Lines 588-589).

XXVI. *Figure 4A: "1,250 cells SC cells were seeded in 3D in type I collagen and incubated for 14 days with azithromycin (10uM), SN-38 (20 nM), or both". The word cells is repeated twice.*

Response: Changed to remove one instance of the word cells. (Line 614).

XXVII. *Figure S2 and Figure S4B: I would include these two figures in the corresponding main figures to have a proper view of the well and the magnified part.*

Response: Whole well images from Fig. S2 were added to the corresponding main figure as suggested (new Fig. 1E). Due to size considerations, whole well images from Fig. S4B were not added to the corresponding main figure but remain available for readers in Fig. S3B.

XXVIII. *Figure S4C-D: which graph corresponds to each antibiotic? Please indicate it*

Response: Antibiotic labels to graphs in this figure have now been added. (Fig. S4C-D)

XXIX. *Figure S5A-B: "(A-B) Comparison of (A) median colony area and (B) percent colonies for additional hits to the plate they were located on for the screen". Consider reformulating the last part of the sentence.*

Response: Last part of this sentence has now been changed as suggested.

XXX. *Figure S5C: With that magnification the lumens are not visible*

Response: Magnified insets to show hollow lumens after drug treatment have now been shared in new Figs. 4C and S4D.

XXXI. *Figure S6B-C: The Authors do not comment anything about them in the main text.*

Response: We have now removed this figure entirely, which contained alternative statistical analysis of patient data shared in Fig. 5D (old Fig. 4C), as it largely reiterated data already presented in a main figure.